# Effect of Extracts from Dominant Forest Floor Species of Clear-Cuts on the Regeneration and Initial Growth of *Pinus sylvestris* L. with Respect to Climate Change

**DOI:** 10.3390/plants10050916

**Published:** 2021-05-02

**Authors:** Vaida Sirgedaitė-Šėžienė, Adas Marčiulynas, Virgilijus Baliuckas

**Affiliations:** 1Laboratory of Forest Plant Biotechnology, Institute of Forestry, Lithuanian Research Centre for Agriculture and Forestry, Liepų str. 1, LT-53101 Girionys, Kaunas District, Lithuania; 2Department of Forest Protection and Game Management, Institute of Forestry, Lithuanian Research Centre for Agriculture and Forestry, Liepų str. 1, LT-53101 Girionys, Kaunas District, Lithuania; adas.marciulynas@lammc.lt; 3Department of Forest Genetics and Tree Breeding, Institute of Forestry, Lithuanian Research Centre for Agriculture and Forestry, Liepų str. 1, LT-53101 Girionys, Kaunas District, Lithuania; virgilijus.baliuckas@lammc.lt

**Keywords:** scots pine, germination, seedlings growth, chlorophyll, carbon dioxide, temperature, allelopathic

## Abstract

Climate change influences the ecological environment and affects the recruitment of plants, in addition to population dynamics, including Scots pine regeneration processes. Therefore, the impact of cover-dominant species extracts on the germination of pine seeds and morpho-physiological traits of seedling under different environmental conditions was evaluated. Increasing temperature reinforces the plant-donor allelochemical effect, reduces Scots pine seed germination, and inhibits seedling morpho-physiological parameters. Conditions unfavourable for the seed germination rate were observed in response to the effect of aqueous extracts of 2-year-old *Vaccinium vitis-ideae* and 1-year-old *Calluna vulgaris* under changing environmental conditions. The lowest radicle length and hypocotyl growth were observed in response to the effect of 1-year-old *C. vulgaris* and 2-year-old *Rumex acetosella* under increasing temperature (+4 °C) conditions. The chlorophyll *a + b* concentration in control seedlings strongly decreased from 0.76 to 0.66 mg g^−1^ (due to current environmental and changing environmental conditions). These factors may reduce the resistance of Scots pine to the effects of dominant species and affect the migration of Scots pine habitats to more favourable environmental conditions.

## 1. Introduction

Climate change, in combination with other factors, influences the ecological environment; subsequently, this clearly affects the recruitment and population dynamics of plants [1]. The early stages of the plant life-cycle are expected to be more sensitive to climate change than the adult stages and, as such, represent a major bottleneck for recruitment [2]. Seedling emergence is usually synchronized with seasonal changes in the environment [3]. Germination of some species occurs soon after dispersal, whereas that of other dormant species is postponed until a favourable season when the seedlings can survive, grow, and ultimately reproduce [4]. Due to the influence of post-germination traits and fitness, germination and possibly phenotypic plasticity on dormancy may be subject to strong selection [5]. Therefore, the pertinent issue is whether the adaptation of seeds and seedling traits can match the rate of climate change [6]. Moreover, such adaptation of annual (or herbaceous) species with rapid life cycles may occur more rapidly than that of perennial (or woody) species [7]. With respect to the plasticity and adaptations of tree species, very little is known about their germination behaviour in response to environmental changes [8,9].

The capacity of plants to adapt to changing climatic and environmental conditions is becoming a pressing issue, and has practical importance. Adaptation capacity differs for various plant species [10]. Climate change can challenge the adaptive capacity of many long-lived species, such as trees, and this adaptive capacity likely exceeds migration capacity [11,12]. Species with a wide distribution are especially useful for testing changes in recruitment patterns as a consequence of climate alterations because they grow under considerably different ecological conditions across their range. Scots pine (*Pinus sylvestris* L.) has the largest geographical distribution among pine species, and it is one of the most widespread conifers on Earth, distributed from the Mediterranean to the Arctic [13]. In the EU member states it constitutes approximately 20% of the commercial forest area and is of considerable importance as a timber-producing species, especially in the north of Europe [14]. 

Allelopathy, competition, soil nutrient imbalance, and poor ectomycorrhization have been implicated in conifer regeneration failure in the presence of dense ericaceous understory resulting from forest harvesting and fire in all ecosystem types. There are reports about allelochemical production in many woody species, such as in eucalyptus forests in Australia [15], in boreal conifer forests [16], tropical forests [17], temperate forests [18] and in subdesert communities [19]. Čaboun [20] noted that allelopathy is an important phenomenon for forest regeneration, population structure, and productivity. In forest ecosystems, allelopathy, which is one of the forms of plant competition, affects many plant characteristics, including abundance, growth, plant successions, plant population structure, dominance, diversity, and plant productivity [16]. It has an impact not only on natural regeneration, but also on reforestation, both for seeds germination and the initial growth of seedlings.

Within the forest floor litter, the dominant species of clear-cuts of Scots pine stands, which are mostly ericaceous, contain a range of secondary metabolites, mostly phenolic compounds, that are inhibitory to conifer seed germination, including primary root growth and ectomycorrhizal growth [16]. This inhibition can include planted tree seedlings that exhibit stunted growth in these forest areas with serious ecological, economic, and social consequences. Tree harvesting and inadequate management of the forest can cause long-term occupancy of a site by the understory and cover grown species with irreversible habitat degradation, which converts conifer forests into ericaceous heath [21]. The literature reports that allelopathic features of dominant species may depend on temperature and CO_2_ concentration changes, which are also associated with climate changes [22,23,24]. In addition, previous studies have shown that plant allelopathic properties and Scots pine responses to different temperature regimes vary considerably [25]. Mittler and Blumwald [26], and Ainsworth and Rogers [27] reported that Scots pine seedlings are more easily adaptable to changes in environmental conditions than to plant-dominant extracts. Other studies indicated pine seed germination and growth responses to elevated CO_2_ concentration [28]. The capacity of trees to capitalize on CO_2_ and convert it into carbohydrates results in better plant growth and yield [29]. This is called the CO_2_ fertilization effect and has received less attention compared with secondary climate change factors, such as increasing temperature [30]. Therefore, climate change scenarios have been modelled to predict the potential impact of dominant species of clear-cuts on natural forest regeneration and reforestation under changing climate conditions (temperature and carbon dioxide concentration).

Laboratory studies of allelopathic effect are affected by the knowledge gap associated with imitating natural conditions to evaluate the same processes of allelopathic phenomena. However, research on this topic is widely performed by creating different conditions and setting different goals to better understand certain aspects of this plant-plant interaction. It is important to recognize that it is sometimes impossible to mimic climate changes during laboratory experiments, as the studied plants are sensitive to indoor cultivation. According to our previous studies and literature analyses, the aqueous extracts of dominant forest-floor species have an allelopathic effect, which may have an impact on the morpho-physiological parameters of Scots pine initial growth. Therefore, this study aimed to evaluate the impacts of aqueous extracts produced from both shoots and roots of 1-year-old and 2-year-old dominant plants grown in clear-cuts of Scots pine (*Vaccinio-Pinetum*) stands in response to different temperature regimes and CO_2_ concentrations *ex situ* (in growing chambers). A complex experiment was designed to reveal the implications and interactions between allelopathic and plant developmental parameters (species and plant parts) in the germination of Scots pine seeds and morpho-physiological traits of seedlings.

## 2. Results

### 2.1. The Impact of Environmental Conditions on Seed Germination and Seedling Growth under the Effect of Plant Donors

Analysis of the impact of different environmental conditions (temperature and carbon dioxide) on Scots pine seed germination showed that the aqueous extracts of plant donors exhibited a higher allelopathic effect under increasing temperature (T—25.5/20 °C, CO_2_—400 ppm) and increasing temperature with higher CO_2_ concentrations (T—25.5/20 °C, CO_2_—700 ppm) (Figure 1).

RDA revealed that the rate of seed germination was the lowest as a response under conditions of increasing temperature with higher CO_2_ concentration (T—25.5/20 °C, CO_20_—700 ppm) (Figure 1). The rate of seed germination of Scots pine was more inhibited by 1-year-old clear-cuts of *C. vulgaris* shoot aqueous extracts than by those of other dominant species. In addition, the increasing temperature with higher CO_2_ concentration was more unfavourable for the rate of seed germination due to the effect of 2-year-old clear-cut shoot aqueous extracts of *V. vitis-idaea* (Figure 1). 

A statistically significant decrease in the germination rate was determined in the effect of 2-year-old *V. vitis-idaea* (*p* < 0.05) and *R. acetosella* (*p* < 0.05), and 1-year-old *C. vulgaris* (*p* < 0.05) shoot extracts when the temperature increased from 21.5/16 °C to 25.5/20 °C and the CO_2_ concentration increased from 400 ppm to 700 ppm (Figure 2).

Statistical analysis (Table 1) revealed the statistically significant inhibition of *C. vulgaris* (*p* < 0.05) extracts and 2-year-old *V. vitis-idaea* (*p* < 0.05) shoot extracts on Scots pine seed germination rates when the temperature increased from 21.5/16 °C to 25.5/20 °C. The statistically significant lowest seed germination rate was recorded in the effect of the shoot extracts of *C. epigejos* (*p* < 0.05) under increasing temperature (25.5/20 °C) and higher (700 ppm) CO_2_ concentration conditions (Table 1).

Under conditions of increasing temperature and higher CO_2_ concentration, different plant donors and their parts exhibited a stronger inhibitory effect on the growth of Scots pine seedling radicles compared with the growth of Scots pine seedling hypocotyls (Figure 3).

According to RDA (Figure 3) analysis, the inhibitory effect on the growth of Scots pine radicles increased with increasing temperature (T—25.5/20 °C, CO_2_—400 ppm). The strongest inhibitory effect on the growth of Scots pine seedling radicles was determined in shoot aqueous extracts of 1-year-old *V. vitis-idaea* and *P. schreberi*, and of 2-year-old *R. acetosella* and *C. epigejos.* The results of our research (Figure 4) indicate that the radicle length was more inhibited in the effect of 1-year-old *C. vulgaris* (*p* < 0.05) and *R. acetosella* (*p* < 0.05) under increasing temperatures from 21.5/16 °C to 25.5/20 °C.

RDA revealed (Figure 5) that the increase in temperature from 21.5/16 °C to 25.5/20 °C significantly reduced the hypocotyl length due to the effect of the aqueous extracts of all plant-donors. In addition, under the increased temperature and higher CO_2_ concentration conditions (T—25.5/20 °C, CO_2_—700 ppm), plant-donor aqueous extracts exhibited lower inhibitory effects on the hypocotyl length of Scots pine (Figure 5).

The hypocotyl length of Scots pine seedling were more inhibited by the effect of shoot aqueous extracts of *C. vulgaris* of 1-year-old clear-cuts and *R. acetosella* of 2-year-old clear cuts than by those of other plant-donor species (Figure 6).

Statistical analysis revealed that the increase in temperature from 21.5/16 °C to 25.5/20 °C (Table 2) significantly reduced the radicle length due to the effect of shoot aqueous extracts of 2-year-old *C. epigejos* (*p* < 0.05) and *R. acetosella* (*p* < 0.05). The radicle length of Scots pine seedlings significantly decreased due to the effect of the shoot aqueous extracts of *C. epigejos* (*p* < 0.05), *R. acetosella* (*p* < 0.05), 1-year-old *V. vitis-idaea* (*p* < 0.05), and *P. schreberi* (*p* < 0.05) under conditions of increasing temperature with higher CO_2_ concentration conditions (T—25.5/20 °C, CO_2_—700 ppm) (Table 2).

The hypocotyl length of Scots pine seedlings significantly decreased due to the effect of shoot aqueous extracts of 2-year-old *R. acetosella* and 1-year-old *C. vulgaris* under conditions of an increase in temperature from 21.5/16 °C to 25.5/20 °C (*p* < 0.001 and *p* < 0.05, respectively), and an increase in temperature with higher (700 ppm) CO_2_ concentration (*p* < 0.01 and *p* < 0.05, respectively) (Table 2).

### 2.2. Allelopathic Effect on Photosynthetic Pigment Content under Different Environmental Conditions

The content of chlorophyll *a* + *b* of Scots pine seedlings was significantly affected by the shoot aqueous extracts of *P. schreberi* (*p* < 0.05), *V. vitis-idaea* (*p* < 0.05), and *C. vulgaris* (*p* < 0.01) under conditions of an increase in temperature from 21.5/16 °C to 25.5/20 °C (Figure 7). A lower inhibitory allelopathic effect on Scots pine seedling chlorophyll *a* + *b* content was determined for all plant-donor root aqueous extracts (Figure 7).

The allelopathic effect of plant-donor extracts under increasing temperature with higher CO_2_ concentration conditions was comparable to that obtained under increasing temperatures.

The Scots pine seedlings not affected by aqueous extracts of the plant-donor produced a higher content of carotenoids due to increasing temperature (Figure 8). Under conditions of increasing temperature with higher CO_2_ concentration, the higher CO_2_ concentration reduced the negative effect of the increasing temperature; therefore, the content of carotenoids was comparable to that obtained in the control, which amounted to 0.15 mg g^−1^ (Figure 8).

Plant-donor root aqueous extracts exhibited a lower negative impact on the carotenoid content of Scots pine seedlings (Figure 8). The highest changes in carotenoid content were determined in the effect of shoot aqueous extracts of *V. vitis-idaea* (*p* < 0.01) and root aqueous extracts of *C. vulgaris* (*p* < 0.01) under conditions of increasing temperature with higher CO_2_ concentration (T—25.5/20 °C, CO2—700 ppm). In addition, the increase in temperature significantly reduced the carotenoid content as a result of shoot aqueous extracts of 1-year-old *P. schreberi* (*p* < 0.01) and *V. vitis-idaea* (*p* < 0.01) and root aqueous extracts of 1-year-old *C. vulgaris* (*p* < 0.01) and 2-year-old *C. epigejos* (*p* < 0.01) (Figure 8).

The highest statistically significant inhibition of the chlorophyll *a* + *b* content in Scots pine seedlings was observed in the effect of shoot aqueous extracts of *P. schreberi* (*p* < 0.01) under conditions of increasing temperature from 21.5/16 °C to 25.5/20 °C with higher CO_2_ concentration (700 ppm) (Table 3).

Statistical analysis revealed the statistically significant inhibition of the chlorophyll *a* + *b* content of Scots pine seedlings due to the effect of root aqueous extracts of 1-year-old *C. vulgaris* (*p* < 0.01) and 2-year-old *V. vitis-idaea* (*p* < 0.01). The chlorophyll a + b content of Scots pine seedlings was more inhibited under conditions of increasing temperature with higher CO_2_ concentration (Table 3).

The strongest statistically significant inhibition of carotenoid concentration in Scots pine seedlings was recorded in the effect of shoot aqueous extracts of 1-year-old *V. vitis-idaea* (*p* < 0.001) and root aqueous extracts of 2-year-old *C. epigejos* (*p* < 0.001) in increasing temperature with higher CO_2_ concentration conditions (Table 3). Shoot aqueous extracts of 2-year-old *C. epigejos* (*p* < 0.01) and 1-year-old *P. schreberi* (*p* < 0.01) significantly changed the carotenoid concentration in Scots pine seedlings. This included root aqueous extracts of 1-year-old *C. vulgaris* (*p* < 0.01) and 2-year-old *V. vitis-idaea* (*p* < 0.01) under conditions of increasing temperature with higher CO_2_ concentration (T—25.5/20 °C, CO_2_—700 ppm) (Table 3).

### 2.3. Interaction between Plant-Donor Species and Environmental Conditions

Variance analysis showed (Table 4) that not only climate conditions but also the interaction between the dominant species and environmental conditions had a significant impact on the germination rate and the carotenoid content.

The radicle length (*p* < 0.01) and hypocotyls (*p* < 0.05) were mostly affected by the interaction between the dominant species and environmental conditions in addition to the pH of the species extracts. The environmental conditions and the dominant species were the primary indicators that most significantly contributed to variability in the obtained estimates. In addition, it was determined that the pH of the species extracts significantly affected the germination rate (*p* < 0.01) and the carotenoid content (*p* < 0.05). A statistically significant low-positive correlation (r = 0.21) was determined between the extract pH and the germination rate. A low positive correlation (r = 0.25) was found between the extract pH and the carotenoid content.

## 3. Discussion

Our findings show that the higher temperature was the main factor that increased the inhibitory effect of plant-donor aqueous extracts on germination of seeds and initial growth of seedlings. However, a study conducted by Ruan et al. [3] observed that the rising environmental temperature could increase the inhibitory effect of the aqueous extracts of *P. schrenkiana* needles on the seed germination and seedling growth of *P. schrenkiana.* At the germination stage, the influence of the roots of dominant plants is more important, because the seed is affected by processes occurring in the soil. Allelochemical produced by a donor plant determines its mobility in water, volatility in air, affinity to soil surfaces, and degradability [31]. 

According to our results, the increasing CO_2_ concentration recused the initial growth of Scots pine seedlings. In addition, the most favourable conditions for the length of hypocotyls were observed at current environmental conditions. Studies conducted by some authors indicated pine seed germination and growth responses to elevated CO_2_ concentration [28]. They indicated contradictory results that the seedings of *Pinus pinea* L. and that *Pinus sylvestris* L. were more sensitive to higher CO_2_ concentrations. Other authors [26,27] reported that Scots pine seedlings are more adaptable to changes in environmental conditions than to plant-dominant extracts. Many C_3_ plant species, particularly trees and C_3_ grasses, can capitalize on CO_2_ and convert it into carbohydrates due to increases in the photosynthesis rate, which results in better growth and yield [29]. Because of this generally positive assessment, which is sometimes called the CO_2_ fertilization effect, CO_2_ itself seems to receive less attention compared with secondary climate-change factors, such as increasing temperature [30]. The positive effects of CO_2_ could be welcomed, or even necessary to offset the anticipated negative impacts of high-temperature stress [22]. Mahmoodzadeh et al. [30] noted that, under unfavourable environmental conditions, different plant donors and their parts exhibited a stronger inhibitory effect on the seedling radicle compared to that on hypocotyl growth in Scots pine stands. Our results indicated that seed germination exhibited a different response to the root and shoot extracts of dominant species under different environmental conditions. Moreover, some authors have indicated that allelochemical production could be genetically regulated [31,32,33], and that allelochemical concentration depends on plant age [34,35]. Our results confirm these findings and indicate significant differences in Scots pine morpho-physiological parameters due to the effect of extracts of *V. vitis-idaea* of different ages. 

The failure of natural regeneration and growth inhibition of planted conifers, such as black spruce, in the presence of *Kalmia angustifolia* L., and *Ledum groenlandicum* L. has been reported by several authors [36,37]. The ericaceous shrubs in coastal oceanic temperate rainforests and several Vaccinium species (e.g., *Vaccinium alaskaense* L.) in high elevation forests have been reported to cause growth stagnation of conifers such as *Tsuga plicata* Donn, *Thuja heterophylla* (Raf. Sarge), *Picea sitchensis*, and *Abies amabilis* Dougl. [38,39,40]. In sub-alpine spruce forests *Vaccinium myrtillus* L. has been reported to cause the regeneration failure of *Picea abies* (L) Karst. [41]. Phenolic allelochemicals of forest floor humus and seed predation have been implicated in this regeneration failure [42,43]. Experiments have been designed to determine the interactions between tree seedlings and ericaceous plants manifested via direct contact of dominant plant roots and tree seedling roots [44,45]. 

The inhibitory effect of different plant-donor part extracts on photosynthetic pigment under different environmental conditions might be due to the imbalance in metabolism regulated by various enzyme activities [46]. The structure of the plasma membrane may become denatured by the allelochemicals present in the shoot extract. The effect of different donor-plant parts has also been indicated in studies by other authors [30,47,48]. The photosynthetic pigments content was mostly affected by increasing CO_2_ concentrations [28]. This process might be related to a decrease in chlorophyll *a/b*-binding proteins, which could reduce the risk of photooxidative damage due to the relative decrease in the absorption cross-section of photosystems [48]. Alexieva et al. [29] noted out that the increase in the content of photosynthetic pigments depended on plant adaptation to changing conditions, that is, a defensive reaction to a stress agent (for example, increasing temperature and CO_2_ concentration). Existing local adaptations that allow individuals to tolerate environmental conditions are especially important to increase the resistance and resilience of current populations. As mentioned previously, the chlorophyll content and the radicle length may be used as key parameters in studies because of their sensitivity to allelopathic compounds [49]. Plant allelochemicals adversely affect chlorophyll biosynthesis and accumulation by interfering in chlorophyll biosynthesis and/or destruction. The subsequent negative effects of these processes inhibit photosynthesis and plant growth [50].

Any alterations in the plant community structure that are caused by climate change result from underlying changes in the population dynamics of the species that make up the community [51]. In an environmental context, it is very difficult to predict the results of climate change on allelopathic processes, but additional synergic or antagonistic interactions between allelochemicals and environmental conditions could be expected during experiments [52]. Therefore, understanding the responses to climate change at the species level is important to predict the functioning of the ecosystem in the future [53]. Species with higher adaptive capacity may remain, but the risk of a loss of genetic diversity could increase. Furthermore, the competitiveness of various plant species may be affected by the changing climate. 

## 4. Materials and Methods

### 4.1. Dominant Species

The study of the successional stages of herbaceous plant and shrub phytocoenoses was conducted in one-year-old and two-year-old clear-cuts after reforestation of *Pinetum vacciniosum* forests in June-July of 2015. In total, seventeen clear-cuts were evaluated, including eight one-year-old and nine two-year-old clear cuts in the Trakai Forest Enterprise in the south-eastern part of Lithuania (54°30′–54°38′ N, 24°50′–25°00′ E). Aiming to determine the dominant species, the frequency (**%**) and relative plant cover (**%**) of ground vegetation species were estimated in 1 m^2^ subplots (*n* = 25) along transect in every 5 m in each site. The assessment was undertaken using the Braun–Blanquet [54] scale. The soil in the site was classified as *Arenosol* [55]. The forest type of the site was *Pinetum vacciniosum*, and the forest site type was Nb (oligotrophic mineral soil of normal moisture), according to the Lithuanian classification [56]. The mean annual temperature in Lithuania is 6.5 °C, and the mean annual precipitation is 686 mm.

During the assessment of dominant species of the experimental plots in the one-year-old clear-cut after reforestation, the highest frequency was found to be of *Pleurozium schreberi* and *Vaccinium vitis-idaea* (97% and 95%, respectively). *Calluna vulgaris* was the third most dominant species in the 1-year-old clear-cut at a frequency of 87%. The relative plant cover of *Vaccinium vitis-idaea* increased by 2%,reached 12%, and maintained its dominant position in relation to other plants in the two-year-old clear-cuts. The highest frequencies in the 2-year-old clear-cut after reforestation were recorded for *Calamagrotis epigejos* and *Rumex acetosella*, of 44% and 68% respectively. These two species were classified as dominant due to their vigorous and various spreading abilities (both generative and vegetative), and these species structurally altered the phytocoenosis that was previously established in the clear-cuts.

Subsequently, the dominant species (plant-donors) were used for production of aqueous extracts for the assessment of their allelochemical impact on potential acceptor species. Seed germination and seedling growth of the acceptor plant (*Pinus sylvestris* L.) were assessed as the main indicators of the morphophysiological traits of dominants on the initial stages of forest regeneration in the clear-cuts.

### 4.2. Laboratory Bioassays

The fresh dominant plants were sampled in the 1-year-old and 2-year-old forest clear-cuts after reforestation at mid-day. Each sample was collected in three replicates of dominant plants from three 1-year-old and three 2-year-old forest clear-cuts. The samples were transported to a laboratory in plastic bags. For the production of 0.2% aqueous extract concentrations, air-dried plant parts (roots and shoots) were chopped into 2 cm sized pieces. Twenty grams of biomass were placed in glass bottles filled with 100 mL of distilled water; then, the samples were covered with foil, shaken several times, maintained for approximately 12 h at 18 to 20 °C, and filtered. This procedure was in a previous study [25].

For the assessment, 100 sterilized Scots pine seeds were obtained from the Forest Seed and Plant Quality Department of the Lithuanian State Forest Service. It should be noted that Scots pine plus trees in Lithuania were selected mainly from the *Pinetum vacciniosum* forest type and soil conditions are similar to the current experiment’s soils. The composition of genotypes in seed orchards is broad in origin, so genetic diversity is also expected to be large. Pine seeds for the experiment were taken from the seed lots of several previous years and a mixture of seed orchards. The applicability of the results is defined by the forest type and soil conditions that actually prevail in pine stands in Lithuania. Pine stands occupy the largest forest area in Lithuania. Scots pine seeds were placed on filter paper in a Petri dish with a 9-cm-diameter. Five milliliters of 0.2% extract were added to each Petri dish and covered. For the control, 100 seeds were sown in distilled water. An additional 5 milliliters of distilled water and aqueous extracts from the dominant plants were added to Petri dishes every five days during the germination period. Each variant was performed in triplicate. The total number of Petri dishes was 324 (2 ages of clear-cuts × 3 replicates of clear-cuts × 3 dominant species × 2 plant parts × 3 different experimental conditions × 3 replicates × 3 replicates of Petri dishes = 324 Petri dishes).

To evaluate the impact of different temperatures and CO_2_ concentrations on the allelopathic activity of dominant plants, the influence of different dominant species and plant parts on the germination and seedling growth of the Scots pine was assessed. The laboratory trials were carried out in the laboratory of Vytautas Magnus University. For the seed germination process, all Petri dishes were stored in chambers with controlled temperatures and CO_2_ conditions that consisted of three different temperatures and CO_2_ concentrations for 21 days. The photoperiod was 14 h, the air humidity was 75%, and the photosynthetic photon flux density was approximately 200 μmol m^2^ s^−1^ in all chambers. Light was provided by the combination of six natural fluorescent lamps (Philips, Waterproof OPK Natural Daylight LF80 Wattage 2 W/TL-D 58 W) and one high-pressure sodium lamp (Philips MASTER GreenPower CG T 600 W). The Petri dishes were exposed to the following temperatures and CO_2_ concentrations:the first three chambers (current environmental conditions)—temperature of 21.5/16 °C at day/night with 400 ppm CO_2_,the second three chambers (increasing temperature (+4 °C) conditions)—25.5/20 °C with 400 ppm CO_2_,the third three chambers (increasing temperature with higher CO_2_ concentration conditions)—25.5/20 °C with 700 ppm CO_2_.

The photoperiod, air humidity, and photosynthetic photon flux density were automatically adjusted. The temperatures and CO_2_ concentrations were controlled by the PC-based Environmental Control System (computer software IGSS 9-13175). 

Measurements of Scots pine seedlings were performed. The germinated seeds were counted (in units) at day 21. Germination rates (G,%) were calculated relative to germination in the control (C) according to the formula [57]:G = (Gv × 100)/C(1)
Gv—is the number of germinated seeds per unit in the experiment, C - is the number of germinated seeds in the control equated to 100%.

At day 21 the radicle and hypocotyl lengths (mm) were measured and deducted as a percentage of control as germination rate.

In addition, 0.2 g of fresh pine seedling shoots was sampled for chlorophyll *a* and *b* (chl *a* and *b*) and carotenoid assays. For the determination of pigment content, the method of Wettstein [58] was used, as described in a previous study [59]. The content of carotenoids and chl *a* and *b* (mg g^-1^) was estimated spectrophotometrically using the T80 UV-VIS spectrophotometer (PG Instruments, Leicestershire, UK) at wavelengths of 441, 662 and 644 nm (*D*), respectively. For the calculation of pigment content, the following formulas were used:(2)$$\mathrm{chl}a=9.784\times D_{662}-0.990\times D_{664}\frac{c\times V}{P\times 1000},$$
(3)$$\mathrm{chl}b=21.426\times D_{644}-4.650\times D_{662}\frac{c\times V}{P\times 1000},$$
(4)$$\mathrm{Carotenoid}=4.695\times D_{440.5}-0.268\times (\mathrm{chl}a+b)\frac{c\times V}{P\times 1000},$$
where *c* is the pigment content (mg g^−1^), *V* is the extract volume (mL), and *P* is the fresh mass of seedling needles (g).

Laboratory analyses were performed at the Institute of Forestry of the Lithuanian Research Centre for Agriculture and Forestry (LAMMC).

### 4.3. Statistical Data Analysis

The significance of the data was assessed based on Wilkin’s λ-test and Fisher’s theoretical criterion (*F*). The results of the allelopathic effects were statistically evaluated using the statistical package STATISTICA from Stat Soft. SAS UNIVARIATE procedure was used to check if residuals followed normal distribution. Based on the results on residual distribution, the Student *t*-test (*t*) or Kolmogorov–Smirnov (KS) test was used in pairwise comparison to determine statistically significant differences (* *p* < 0.05, ** *p* < 0.01, *** *p* < 0.001) in germination rates and morphophysiological parameters under different temperature regimes (analysed using the XLSTAT software). The Kruskal–Wallis test was used for a group of treatments. Redundancy analysis (RDA) was used to determine relationships between different environmental conditions and the effects of dominant species and their parts on Scots pine germination rate and morphometric parameters (analysed using XLSTAT software). The SAS MIXED procedure (REML method) was used to obtain variance components of random effects and significances of fixed effects. The effects of fixed parameters, that is, temperature and the extract pH group (if 2.0 < pH ≤ 4.9, then the pH group = 1; if 4.9 < pH ≤ 5.9, then the pH group = 2; and if 5.9 < pH < 7.0, then pH group = 3) were evaluated using the TEST3 method.

The initial SAS MIXED procedure was performed according to the following model.
(5)$$y_{ijklmn}=\mu +s_{i}+p_{j}+h_{k}+r_{l}+d_{m}+rh_{lk}+dh_{mk}+rs_{li}+ds_{mi}+\varepsilon_{ijklmn},$$
where *y_ijklmn_* is the response value for the n-th observation, *µ* is the overall mean, *s_i_* is the fixed effect for the *i*-th temperature, *p_j_* is the fixed effect for the *j*-th replication, *h_k_* is the fixed effect for the *k*-th pH group, *r_l_* is the random effect for the l-th species, *d_m_* is the random effect for the *m*-th plant part, *rh_lk_* is the random effect for the interaction of the *l*-th species and the *k*-th pH group, *dh_mk_* is the random effect for the *m*-th plant part and *k*-pH group interaction, *rs_li_* is the random effect for the *l*-th species and *i*-th temperature interaction, *ds_mi_* is the random effect for the *m*-th plant part, the *i*-th temperature interaction, and *ε_ijklmn_* is the random error.Furthermore, the study included a series of models with fixed effects from model (4) but with a single random effect. Therefore, we arrived at six models associated with respective random effects. This part of the statistical analysis was performed using the SAS software (SAS 9.4) [60].

## 5. Conclusions

The assessment of climate conditions revealed the inhibitive properties of dominant extracts on seed germination under changing environmental conditions ex situ (a temperature of 25.5 °C during the day and 20 °C at night and 700 ppm of CO_2_). However, the morphometric parameters of the seedlings remained unchanged. Higher air temperature resulted in a stronger negative effect, which decreased pine germination and seedling growth. The CO_2_ concentration of 700 ppm induced better growth of pine seedlings but decreased the concentration of photosynthetic pigments. The main effect on the germination of Scots pine seeds and morpho-physiological parameters of seedlings was induced by different environmental conditions and interactions between plant-donor species and climatic conditions. 

It can be assumed that the new findings obtained regarding the changes in the allelopathic potential of dominant species in the clear-cuts of previous Scots pine stands in Lithuania will improve the understanding of the impact of allelopathy on reforestation and the further management of forest ecosystems under changing climate conditions. 

## Figures and Tables

**Figure 1 plants-10-00916-f001:**
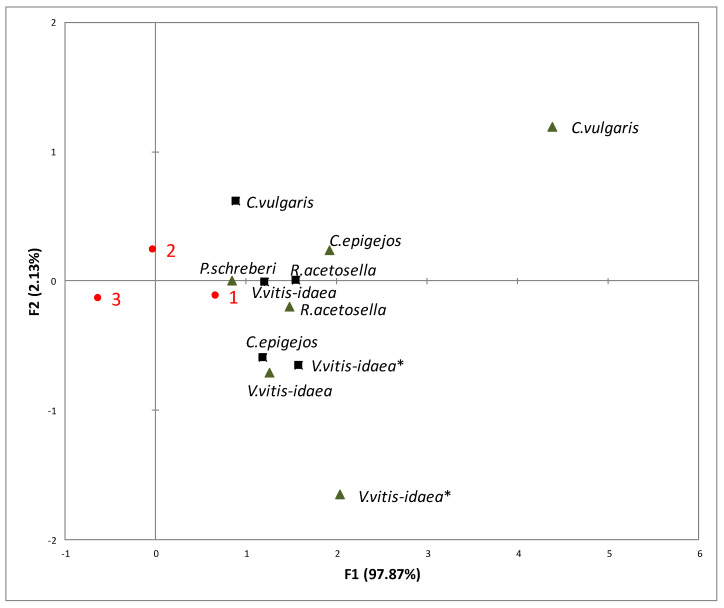
RDA of the impact of the shoots (black triangles) and roots (black squares) of plant-donor aqueous extracts on germination at different climate conditions (1—Current environmental conditions (21.5/16 °C and 400 ppm CO_2_ concentration), 2—temperature increases of 4 °C compared to the current conditions (25.5/20 °C and 400 ppm CO_2_ concentration), 3—Increasing temperature with higher CO_2_ concentration conditions (25.5/20 °C and 700 ppm CO_2_). (*): dominant species of 2-year-old clear-cuts.

**Figure 2 plants-10-00916-f002:**
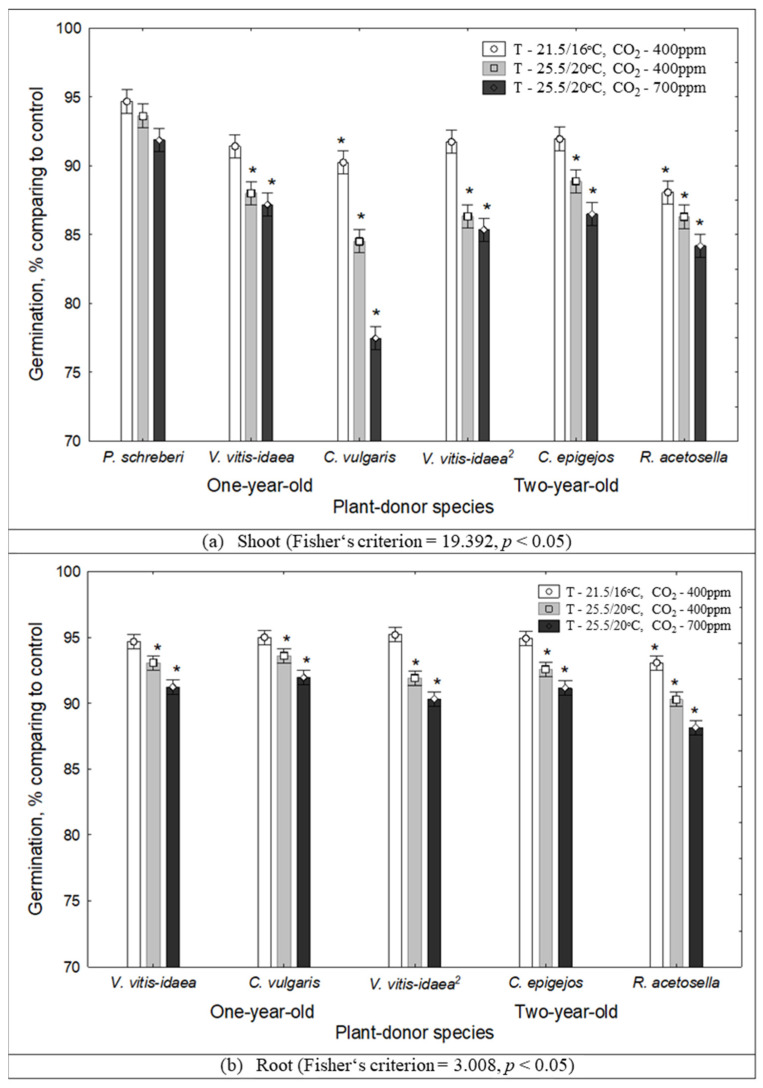
The impact of climate conditions on Scots pine seed germination rate under the effect of plant-donor shoot (**a**) and root (**b**) extracts (mean±SD, *—*p* < 0.05 post-hoc LSD test). Control conditions (21.5/16 °C and 400 ppm CO_2_ concentration), temperature increases of 4 °C compared to the current conditions (25.5/20 °C and 400 ppm CO_2_ concentration), increasing temperature with higher CO_2_ concentration conditions (25.5/20°C and 700 ppm CO_2_). (^2^) - dominant species of 2-year-old clear-cuts.

**Figure 3 plants-10-00916-f003:**
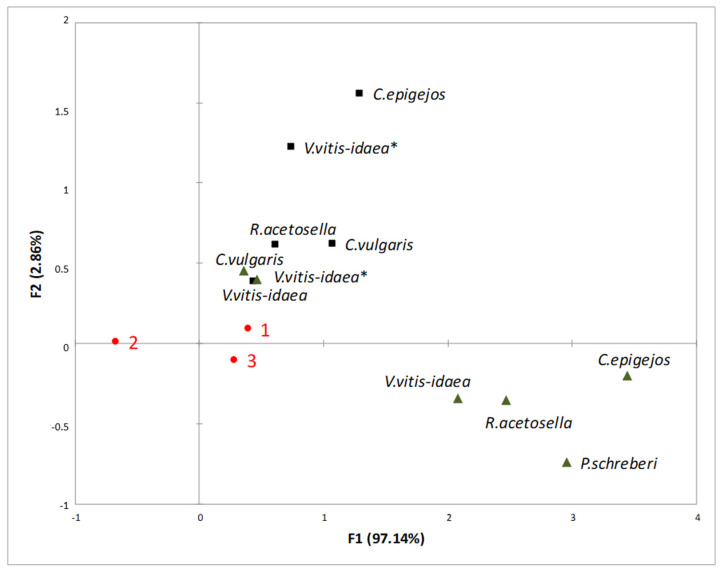
RDA of the impact of the shoots (black triangles) and roots (black squares) of plant-donor aqueous extracts on radicle length at different climate conditions (1—Current environmental conditions (21.5/16 °C and 400 ppm CO_2_ concentration), 2—temperature increases of 4 °C compared to the current conditions (25.5/20 °C and 400 ppm CO_2_ concentration), 3—increasing temperature with higher CO_2_ concentration conditions (25.5/20 °C and 700 ppm CO_2_). (*): dominant species of 2-year-old clear-cuts.

**Figure 4 plants-10-00916-f004:**
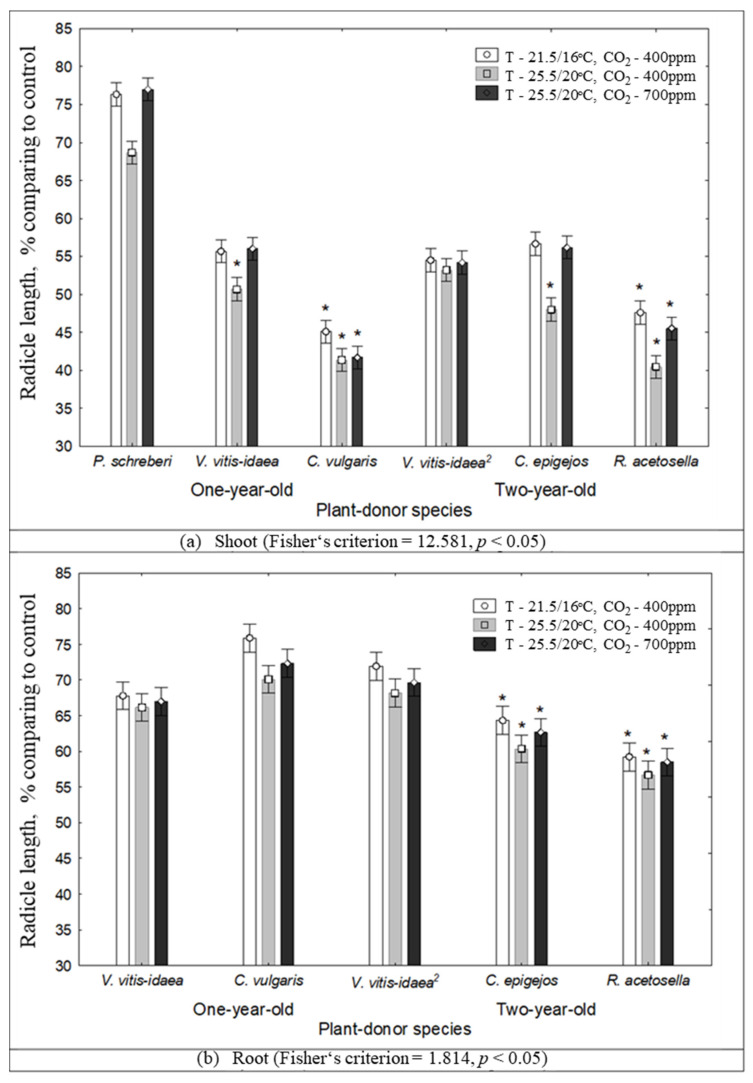
The impact of climate conditions on Scots pine radicle length under the effect of plant-donor shoot (**a**) and root (**b**) extracts (mean±SD, *—*p* < 0.05 Kruskal–Wallis test). Control conditions (21.5/16 °C and 400 ppm CO_2_ concentration), temperature increases of 4 °C compared to the current conditions (25.5/20 °C and 400 ppm CO_2_ concentration), increasing temperature with higher CO_2_ concentration conditions (25.5/20 °C and 700 ppm CO_2_). (^2^)—dominant species of 2-year-old clear-cuts.

**Figure 5 plants-10-00916-f005:**
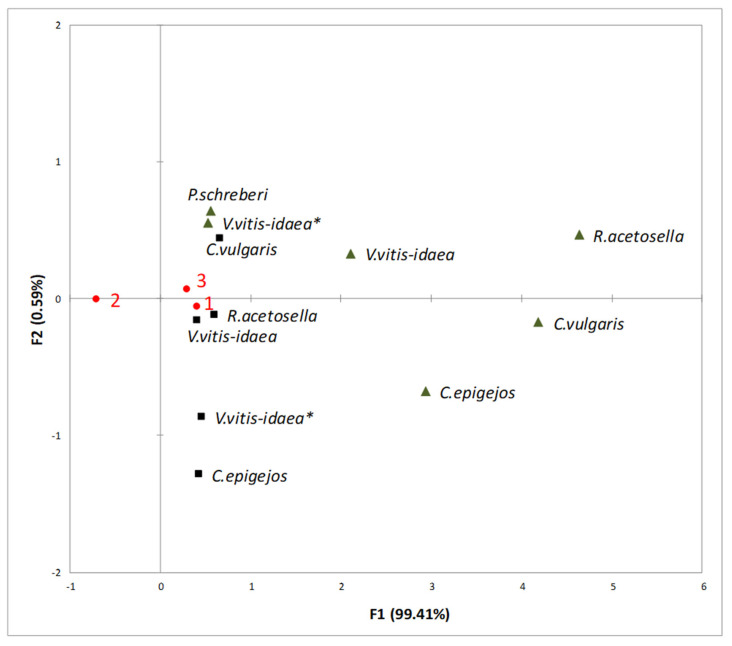
RDA of the impact of the shoots (black triangles) and roots (black squares) of plant-donor aqueous extracts on hypocotyl length at different climate conditions (1—Current environmental conditions (21.5/16 °C and 400 ppm CO_2_ concentration), 2—temperature increases of 4 °C compared to the current conditions (25.5/20 °C and 400 ppm CO_2_ concentration), 3—increasing temperature with higher CO_2_ concentration conditions (25.5/20 °C and 700 ppm CO_2_). (*): dominant species of 2-year-old clear-cuts.

**Figure 6 plants-10-00916-f006:**
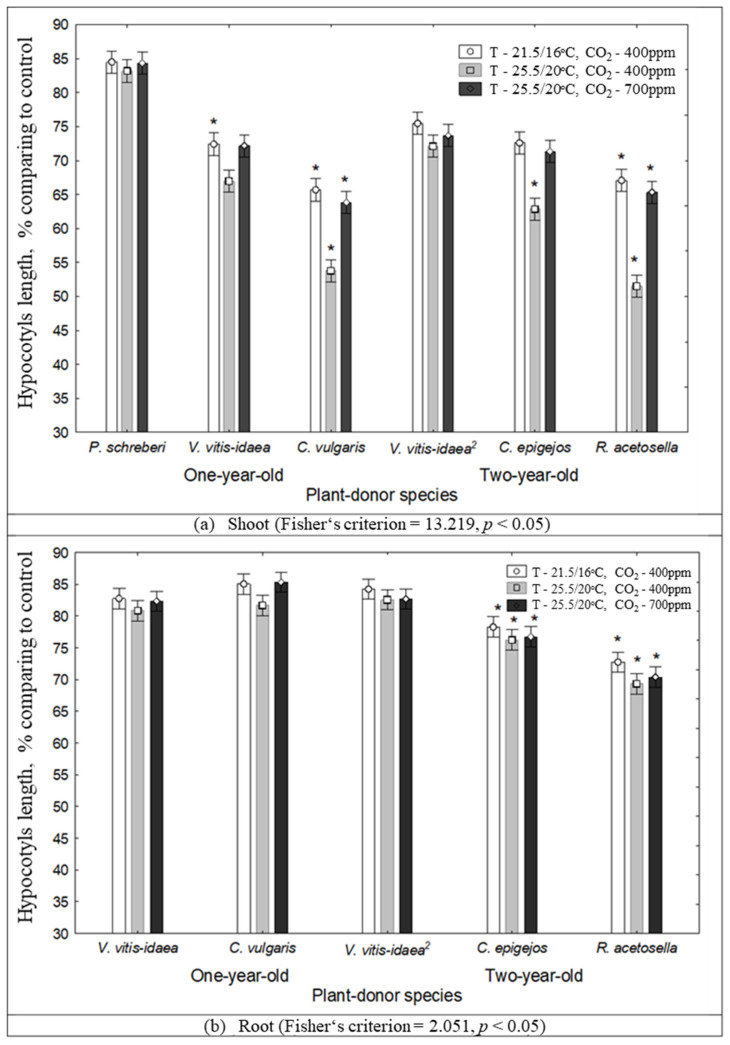
The impact of climate conditions on Scots pine hypocotyls length under the effect of plant-donor shoot (**a**) and root (**b**) extracts (mean ± SD, *—*p* < 0.05 Kruskal–Wallis test). Control conditions (21.5/16 °C and 400 ppm CO_2_ concentration), temperature increases of 4 °C compared to the current conditions (25.5/20 °C and 400 ppm CO_2_ concentration), increasing temperature with higher CO_2_ concentration conditions (25.5/20 °C and 700 ppm CO_2_). (^2^)—dominant species of 2-year-old clear-cuts.

**Figure 7 plants-10-00916-f007:**
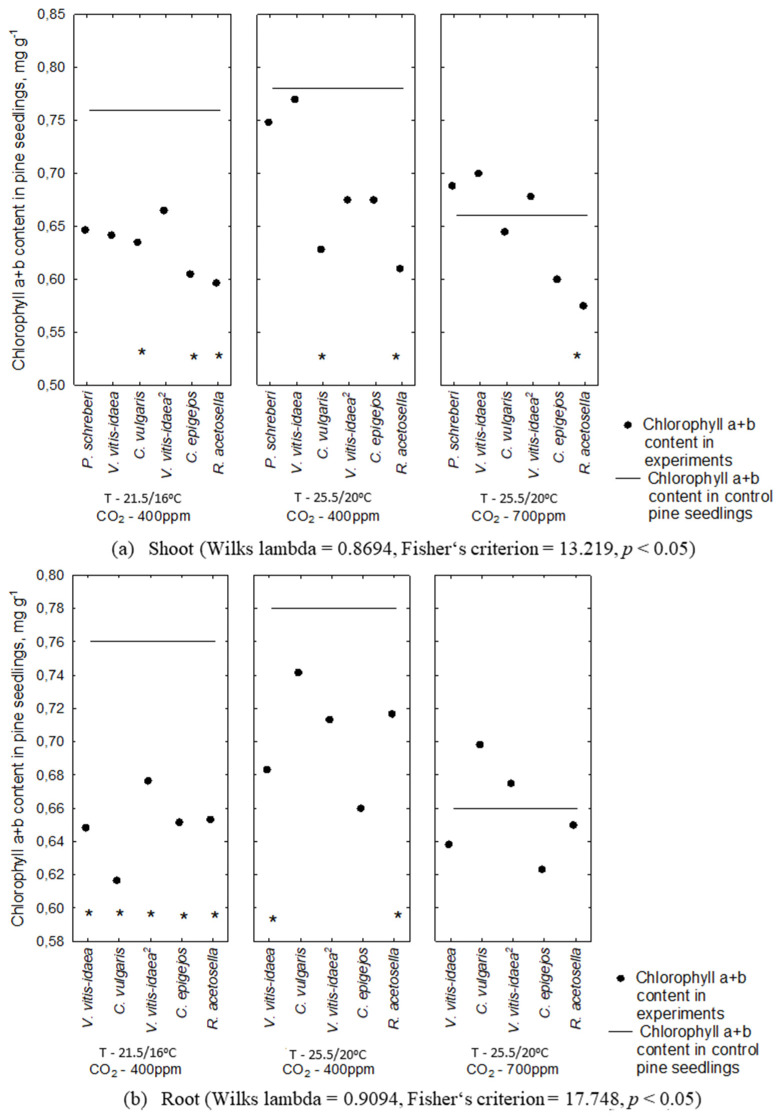
Chlorophyll concentration in the Scots pine seedlings under the effect of plant-donor shoot (**a**) and root (**b**) extracts under different CO_2_ and temperature conditions (*—*p* < 0.05 post-hoc LSD test). Control conditions (21.5/16 °C and 400 ppm CO_2_ concentration), temperature increases of 4 °C compared to the current conditions (25.5/20 °C and 400 ppm CO_2_ concentration), increasing temperature with higher CO_2_ concentration conditions (25.5/20 °C and 700 ppm CO_2_). (^2^)—dominant species of 2-year-old clear-cuts.

**Figure 8 plants-10-00916-f008:**
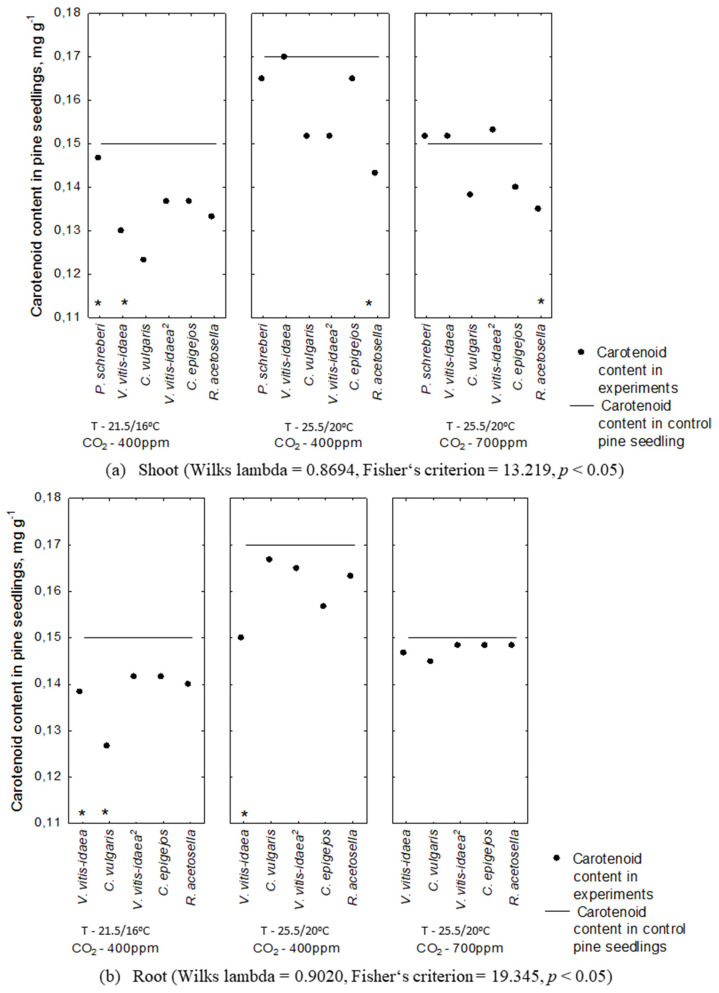
Carotenoid concentration in the Scots pine seedlings under the effect of plant-donor shoot (**a**) and root (**b**) extracts under different CO_2_ and temperature conditions (*—*p* < 0.05 post-hoc LSD test). Control conditions (21.5/16 °C and 400 ppm CO_2_ concentration), temperature increases of 4 °C compared to the current conditions (25.5/20 °C and 400 ppm CO_2_ concentration), increasing temperature with higher CO_2_ concentration conditions (25.5/20 °C and 700 ppm CO_2_). (^2^)—dominant species of 2-year-old clear-cuts.

**Table 1 plants-10-00916-t001:** Overall difference between treatment groups and between the treatments (*n* = 9) for germination rate under different climate conditions (* *p* < 0.05, ** *p* < 0.01, *** *p* < 0.001).

Age of Clear-Cuts	Extract	Plant-Donor	Pr > Chi-Square	Climate Conditions		
				1/3 ^1^	1/2	3/2
1-year-old	Shoot	*P. schreberi*	0.2380			
		*V. vitis-idaea*	0.0156			
		*C. vulgaris*	0.0066	**	*	
	Root	*V. vitis-idaea*	0.0581			
		*C. vulgaris*	0.1275			
2-year-old	Shoot	*V. vitis-idaea*	0.0007	***	*	
		*C. epigejos*	0.0012	*		*
		*R. acetosella*	0.0144	*		
	Root	*V. vitis-idaea*	0.0002	**		
		*C. epigejos*	0.0668			
		*R. acetosella*	0.0147	**		

^1^ 1—Current environmental conditions (21.5/16 °C and 400 ppm CO_2_ concentration), 2—Temperature increases of 4 °C compared to the current conditions (25.5/20 °C and 400 ppm CO_2_ concentration), 3—Increasing temperature with higher CO_2_ concentration conditions (25.5/20 °C and 700 ppm CO_2_).

**Table 2 plants-10-00916-t002:** Overall difference between treatment groups and between the treatments (*n* = 9) for radicle and hypocotyl length (mm) responses to extracts under different climate conditions (* *p* < 0.05, ** *p* < 0.01, *** *p* < 0.001).

Age of Clear-Cuts	Extract	Plant-Donor	Pr > Chi-Square		Comparison of Treatments					
					1/3 ^1^	1/2	3/2	1/3	1/2	3/2
			Radicle length	Hypocotyl length	Radicle length			Hypocotyl length		
1-year-old	Shoot	*P. schreberi*	0.0477	0.8503			*			
		*V. vitis-idaea*	0.8441	0.4291			*			
		*C. vulgaris*	0.9333	0.0903						
	Root	*V. vitis-idaea*	0.8630	0.8708						
		*C. vulgaris*	0.8179	0.6439						
2-year-old	Shoot	*V. vitis-idaea*	0.2528	0.9828						
		*C. epigejos*	0.1589	0.6139						
		*R. acetosella*	0.0497	0.0012		*	*		***	**
	Root	*V. vitis-idaea*	0.9521	0.8082						
		*C. epigejos*	0.8655	0.8391						
		*R. acetosella*	0.9554	0.9176						

^1^ 1—Current environmental conditions (21.5/16 °C and 400 ppm CO_2_ concentration), 2—Temperature increases of 4 °C compared to the current conditions (25.5/20 °C and 400 ppm CO_2_ concentration), 3—Increasing temperature with higher CO_2_ concentration conditions (25.5/20 °C and 700 ppm CO_2_).

**Table 3 plants-10-00916-t003:** Overall difference between treatment groups and between the treatments (*n* = 9) for chlorophyll (mg g^−1^) and carotenoid (mg g^−1^) concentrations under different climate conditions (* *p <* 0.05, ** *p <* 0.01, *** *p <* 0.001).

Age of Clear-Cuts	Extract	Plant-Donor	Pr > Chi-Square		Comparison of Treatments					
					1/3 ^1^	1/2	3/2	1/3 ^1^	1/2	3/2
			Chlorophyll *a + b*	Carotenoid	Chlorophyll *a + b*			Carotenoid		
1-year-old	Shoot	*P. schreberi*	0.0008	0.0008		**	**		**	**
		*V. vitis-idaea*	0.0042	<.0001	*	*		**	***	
		*C. vulgaris*	0.7602	0.0150					*	
	Root	*V. vitis-idaea*	0.0950	0.0638						
		*C. vulgaris*	0.0003	<.0001	**	**		**	***	**
2-year-old	Shoot	*V. vitis-idaea*	0.7241	0.0414				*		
		*C. epigejos*	0.0599	0.0003					**	**
		*R. acetosella*	0.8345	0.6707						
	Root	*V. vitis-idaea*	0.0095	0.0010			**		**	**
		*C. epigejos*	0.2013	0.0495						
		*R. acetosella*	0.0837	0.0108					*	

^1^ 1—Current environmental conditions (21.5/16 °C and 400 ppm CO_2_ concentration), 2—Temperature increases of 4 °C compared to the current conditions (25.5/20 °C and 400 ppm CO_2_ concentration), 3—Increasing temperature with higher CO_2_ concentration conditions (25.5/20 °C and 700 ppm CO_2_).

**Table 4 plants-10-00916-t004:** Variance components and standard errors of random effects as well as the *F* criterion and significance (*p*-value) of fixed effects from separate analyses for each trait (* *p* < 0.05, ** *p* < 0.01, *** *p* < 0.001).

Parameter	Variance Component of Interaction between Dominant Species and Climate Conditions (%±SE)		Pr > Z	Climate Conditions		pH	
				*F* criterion	*p*-value	*F* criterion	*p*-value
Germination	17.42	±9.57	*	7.41	**	7.01	***
Radicle length	38.67	±17.06	**	0.33	-	13.03	***
Hypocotyl length	22.49	±11.14	*	0.87	-	8.08	***
Carotenoid	9.17	±5.16	*	12.23	***	3.89	*

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
