# Peer review of "Effect of Extracts from Dominant Forest Floor Species of Clear-Cuts on the Regeneration and Initial Growth of Pinus sylvestris L. with Respect to Climate Change"

_plants, 2021, doi:10.3390/plants10050916_

Round 1

Reviewer 1 Report

Good work with the revision! My main concerns have been addressed. The authors did a good job with the corrections. I believe that the manuscript is suitable for publication. Congratulations!

Author Response

Dear reviewer

We are very grateful for the valuable additional comments of reviewers for our manuscript “Effect of extracts from dominant forest floor species of clear-cuts on the regeneration and initial growth of Pinus sylvestris L. with respect to climate change”. We have changed the manuscript according to additional remarks of the other reviewers and we have uploaded the revised manuscript. We hope very much that the performed changes improved the manuscript and the revised version will be accepted for publishing in the special issue of the journal “Plants”.  

Sincerely,

Vaida Sirgedaitė-Šėžienė, Ph.D

Corresponding author: e-mail: [email protected]

Laboratory of Forest Plant Biotechnology

Forest Research Institute, Lithuanian Research Centre for Agriculture and Forestry

Reviewer 2 Report

Although the experimental design is not the optimal (it is not possible in all cases to know if treatment effects are caused by the germination temperature, the other environmental factors or their interactions), I feel that the manuscript has improved from the first review round.

The information about the seed origin is still not perfect (the simple statement that it is seed orchard seed is not the level of detail I would hope) the M&M chapter has improved. I still wish that the applicability of the results to all P. sylvestris populations would be discussed better as the material is limited. 

With these flaws the manuscript is on the verge of being of rejection. As the problems are a part of the experimental design, I do not feel there is much to do to improve the manuscript. I must leave the decision of rejection or accepting with minor changes to the Editorial office. Is this the level of scientific work you accept for your journal or not? 

Author Response

Dear reviewer

We are very grateful for the valuable additional comments of reviewers for our manuscript “Effect of extracts from dominant forest floor species of clear-cuts on the regeneration and initial growth of Pinus sylvestris L. with respect to climate change”. We have changed the manuscript according to additional remarks of the reviewers and we have uploaded the revised manuscript. Information on all performed modifications are provided in detail with answers to the remarks of reviewers. We hope very much that the performed changes improved the manuscript and the revised version will be accepted for publishing in the special issue of the journal “Plants”.  

Answers to reviewer  2:

1) The information about the seed origin is still not perfect (the simple statement that it is seed orchard seed is not the level of detail I would hope) the M&M chapter has improved. I still wish that the applicability of the results to all P. sylvestris populations would be discussed better as the material is limited. I still wish that the applicability of the results to all P. sylvestris populations would be discussed better as the material is limited. 

 Reply: According to reviewer suggestions the actual information was added to M&M: It should be noted that Scots pine plus trees in Lithuania were selected mainly from the Pinetum vacciniosum forest type and soil conditions are similar to the current experiment soils. The composition of genotypes in seed orchards is broad in origin, so genetic diversity is also expected to be large. Pine seeds for the experiment were taken from the seed lots of several previous years and a mixture of seed orchards. The applicability of the results is defined by the forest type and soil conditions that actually prevail in pine stands in Lithuania. Pine stands occupy the largest forest area in Lithuania.“ (line 501-508, p. 19)

Sincerely,

Vaida Sirgedaitė-Šėžienė, Ph.D

Corresponding author: e-mail: [email protected]

Laboratory of Forest Plant Biotechnology

Forest Research Institute, Lithuanian Research Centre for Agriculture and Forestry

Reviewer 3 Report

The authors made significant changes to their previous versions, and seriously improved their manuscript. I think that the major questions highlighted by the three reviewers were at least patially corrected. However, there are still some important concerns :

  • english expression  : a significant number of the new sentences in the introduction are not understandable, contains mistakes, or lack words. I strongly suggest a careful reading and correction by a english native speaker (and I'm not one, but I can detect bad english). I will attach a pdf version of your corrected manuscript where I underline in yellow the sentences that should be clarified and improved.
  • a lot of typo mistakes  : in legend of figure 4 and 6 for instance : the lenght of radicle  should be replaced by radicle length

Author Response

Dear reviewer

We are very grateful for the valuable additional comments of reviewers for our manuscript “Effect of extracts from dominant forest floor species of clear-cuts on the regeneration and initial growth of Pinus sylvestris L. with respect to climate change”. We have changed the manuscript according to additional remarks of the reviewers and we have uploaded the revised manuscript. Information on all performed modifications are provided in detail with answers to the remarks of reviewers. We hope very much that the performed changes improved the manuscript and the revised version will be accepted for publishing in the special issue of the journal “Plants”.  

Answers to reviewer  3:

1) English expression: a significant number of the new sentences in the introduction are not understandable, contains mistakes, or lack words. I strongly suggest a careful reading and correction by a english native speaker (and I'm not one, but I can detect bad english). I will attach a pdf version of your corrected manuscript where I underline in yellow the sentences that should be clarified and improved.

Reply: We agree with the reviewer comments, therefore the manuscript was edited for English language by MDPI English Editing Services. All underlined sentences clarified and improved (see in the manuscript). Thank you for critical review, that allowed us to improve our manuscript as much as possible.

2) A lot of typo mistakes  : in legend of figure 4 and 6 for instance : the lenght of radicle  should be replaced by radicle length ! 

Reply: The legend of Figure 4 and 6 changed according reviewer comments in the manuscript (p. 9 and p. 11).

Sincerely,

Vaida Sirgedaitė-Šėžienė, Ph.D

Corresponding author: e-mail: [email protected]

Laboratory of Forest Plant Biotechnology

Forest Research Institute, Lithuanian Research Centre for Agriculture and Forestry

This manuscript is a resubmission of an earlier submission. The following is a list of the peer review reports and author responses from that submission.

Round 1

Reviewer 1 Report

General comments

Review Plants-1110081 – “Effect of extracts from dominants of clear-cuts on the regeneration and initial growth of Pinus sylvestris L. with respect to climate change” by irgedaitė-Šėžienė et al.

This paper reports results on the impact of cover-dominant species extracts on the germination of Pinus sylvestris seeds under different environmental conditions. The Authors presented original research. The study states how the research fills the identified knowledge gap. The structure conforms to the Plants standards.

However, the Introduction showing the context need some improvement. In the Introduction section the Authors should corrected the information about allelopathy phenomenon. I also think it is especially important to correct the first paragraph of the Introduction section as there are some mistakes.

I have also some doubts about the figures. I believe that the font should be increased because it is illegible. It should also be better explained in the text and in the chart what is meant by double notation of a V. vitis-idaea.

In conclusion, I believe that the manuscript fits into the Journal’s aims and scope, and I think it is interesting enough to be published after a minor revision.  

Specific comments:

L19: Chlorophyll a+b -> Chlorophyll a+b (italics)

L31-34: “The atmospheric CO2concentration amounted to a mere 280 ± 10 ppm before the industrial era. The main reason for the increase in CO2concentration to the present levels, is the combustion of fossil fuels. The main reason for the increase inCO2 concentration is the combustion of fossil fuels, that is, the atmospheric CO2 concentration amounted to 280 ± 10 ppm before the industrial era.” – please correct this paragraph!

L35: The Intergovernmental Panel on Climate Change [2] - please distinguish between citation 1 and 2 as it may appear to be the same item.

L70: “This factor, called allelopathy” - Please re-edit this sentence, because the phenomenon of allelopathy occurs not only in stressful situations and is not related only to the production of phenolic compounds.

Figure 1 - please enlarge the fonts in the chart and the x-y-axis captions as they are illegible

Figure 2 - I don't understand why V. vitis-idaea is signed twice in the chart and why one of the species has an asterisk. Please explain this in the figure caption or in the legend.

Figure 3 - please enlarge the fonts in the chart and the x-y-axis captions as they are illegible

Figure 4 - I don't understand why V. vitis-idaea is signed twice in the chart and why one of the species has an asterisk. Please explain this in the figure caption or in the legend.

Figure 5 - please enlarge the fonts in the chart and the x-y-axis captions as they are illegible

Figure 6 - I don't understand why V. vitis-idaea is signed twice in the chart and why one of the species has an asterisk. Please explain this in the figure caption or in the legend.

L458-465 - please correct the font.

Reviewer 2 Report

Manuscript ID: plants-1110081

Effect of extracts from dominants of clear-cuts on the regeneration and initial growth of Pinus sylvestris L. with respect to climate change

General comments

The topic of the study – how will climate change affect the recruitment of Scots pine in relation to competing vegetation – is interesting and worth studying. The language of the manuscript is satisfactory as are the methods chosen for the statistical analysis.

There are, however, two major issues I find troubling. If I understand correctly the idea is to study how increasing temperatures and CO2 concentrations affect the allelopathic effect of common forest floor plants. It can be reasonably hypothesized that the production of secondary metabolites and their excretion will increase due to climate change. To test this hypothesis my thinking would be to subject the donor plants to elevated temperature and CO2 and see is there an increase in secondary metabolite concentrations. This is however not done, but it is tested if the same concentrations would have different impact to germination in slightly different germination environments. I’m not sure if this makes sense.

Again, if I understand correctly, the germination of Scots pine under raised temperature and elevated CO2 concentration is compared only to control (= no allelopathic substances and no altered conditions). Thus, we can see that these extracts, elevated temperature and CO2 all decrease germination, but from the data we do not know are there any interactions. To test the interaction effects we would need not only the applied treatments but also controls with elevated temp and CO2 but no plant extracts. With the current data set we do not know if the second treatment (plant extracts + 25,5/20 °C 400 ppm) decreases germination due to unfavorably high temperature of due to an interaction.

This results in the fact that the discussion and conclusions concerning interactions are unfounded. The inhibitory effect of the plant extracts and the effects of elevated CO2 to the morphology and chlorophyll content are interesting results, but to me the design of the experiment is flawed to meet the aim of the study.

Some specific comments

Line 2: “Dominants of clear-cuts” is a bit cryptic to me. Mention that you are talking about dominant forest floor species.

L30: “Have risen” -> Rose

L67: Mention the economic importance of P. sylvestris?

L70: I don't think all allelopathic compounds are phenolic.

L95-100: Mention where the study is conducted. As you say, P. sylvestris is a widely distributed tree. The results don’t probably apply to all forest types where it grows.

L378-379: What does after reforestation mean? After planting?

L387-389: ISTA rules require test of 400 seeds. You use 100. Either leave out the ISTA reference or point out the difference.

L390: Please give more detail about the seeds you used. Where they from a natural stand or seed orchard? How old was the seed lot? In discussion it should be stated that using just one seed lot in the study is a limitation.

L416-418 and L423-424: I don’t really understand what was done here. Did you follow ISTA protocol and remove seeds from the dishes after germination (4x the length of the seed coat) or measure all the germinated seeds at 21 days?

Reviewer 3 Report

The manuscript entitled “Effect of extracts from dominants of clear-cuts on the regeneration and initial growth of Pinus sylvestris L. with respect to climate change” described the influence of T and [CO2]  increases on the magnitude of allelopathic effects of some understory species on Pinus sylvestris germination and early growth. This experiment is the continuation of a previous one, published in 2019 using the same experimental design : allelopathic effects are measured by submitting Pinus seeds and seedlings to extracts obtained by maceration of 6 plants (root and shoot, 2cm -size) in distilled water. Plants originated from forest clear cuts (one-year and two-year old). Germination and early growth experiments are conducted under three conditions : the “current” one, one with an increase of 4°C, and one with increase of 4°C and of [CO2] (from 400 to 700 ppm). In summary, temperature rise increases the magnitude of allelopathic affects for some donor plant, and some Pinus traits (pigment amount) are more severely affect under changing environmental conditions.

Globally, experiment albeit limited t is of interest, and give results that could enhance our knowledge about plant-plant interactions in a context of global warming but I’ve got several concerns about the paper.

My main and major concern regards the general objective of this experiment which aim to quantify the effect of rising T° and  [CO2]  on allelopathic interactions in forest (clear-cut context). But the authors only take in account the effect of such global changes on the target plant when exposed to elevated T° and  [CO2] . But what about the effect of such conditions on the donor plant metabolism, and then on the final concentration of allelochemicals in leachates and soil (that is replaced in such experiment by plant extract, with all the well known associated bias) ? Of course, it is impossible to mimic such changes, because the studied plants are very delicate to cultivate indoor, but the authors could have imagine to use field system to increase T and CO2  (Open Top Chamber for instance). At least, even they could not use such system, they might discuss this severe limitation to their study, because literature is abundant regarding the effect of global changes on (phenolic) metabolism in the understorey species they use. They should also mention that such changes will also modifiy soil properties and biological activity, that is likely to impact expression of allelopathic potential in the field.

Second, the manuscript, especially the introduction is badly  organized. The beginning that describe global changes is too long and general. Why not emphasized changes that will affect the North Hemisphere, and its greater susceptibility to global changes, that is now well documented ? And in my opinion, this introduction should begin by the regeneration problem of northern forest, and the specific problem links to clear-cut management. Curiously, they are no mention of the clear-cut ecology in the current introduction.

Third, there are global lacks of information on the experimental procedure and ecological status of the donor plants (moss, ericaceous…). Even some experimental details are found in the previous paper (n° 23), and due to the controversial debates about allelopathic experimental evidences, more precise information need to be explicitly mentioned in the Materials and methods :

  • What is the exact procedure for plant collection ? How many plants in each type of clear cut ? and how many replicates by clear-cut, on what surface? Why in the fig 2, no mention of the effect of the subterranean part of the moss ? Are shoot and root the correct term for the moss Pleurozium ? Are the shoot and root coming from the same individual ? For Vaccinium, did shoot mean stems + leaves ?? Sampling rrots from ericaceous is especially tricky : did you collect fine roots also ?
  • At what T° were the samples dried
  • 20 % is quite a high concentration for plant extracts. Please justify this choice
  • How many Petri dish did you finally obtain ?
  • Line 425 : Did pine seedling shoot mean shoot and the emerging needles ? How old are those seedlings ? Are they already green ?
  • Line 453-464 : why mentioning extract pH group ? I’m not sure that the statistical way to analyse germination rate is accurrate, as each Petri dish is independent from the ones that are received the same extracts ?

Finally, there’s lack of literature regarding boreal forest and Pinus sylvestris regeneration, the clear-cut specificity, and some seminal works about importance of allelopathy in regeneration problem : see for instance works of Nilsson, Jaderlund.. The authors seem to consider the understory plants as an homogeneous and "herbaceous"  group (line 349), that is inaccurate for Vaccinium species. The authors stay very vague in their final discussion, with no reference on phylogenetic and phytochemical differences between the donor species (Moss, Ericaceous, Poaceae, Polygonaceae…). By doing this, they miss many points able to strengthen their interpretation.

And a lot of minor inadequate terms, typo errors must be corrected